# Molecular Biology of Pediatric and Adult Male Germ Cell Tumors

**DOI:** 10.3390/cancers13102349

**Published:** 2021-05-13

**Authors:** Mariana Tomazini Pinto, Flavio Mavignier Cárcano, Ana Glenda Santarosa Vieira, Eduardo Ramos Martins Cabral, Luiz Fernando Lopes

**Affiliations:** 1Molecular Oncology Research Center, Barretos Cancer Hospital, Barretos 14784400, Brazil; mariana.tomazini@hcancerbarretos.com.br (M.T.P.); carcano.fm@facisb.edu.br (F.M.C.); eduardo.cabral@hospitaldeamor.com.br (E.R.M.C.); 2Brazilian Childhood Germ Cell Tumor Study Group, The Brazilian Pediatric Oncology Society (SOBOPE), Barretos 14784400, Brazil; anaglenda.santarosa@hospitaldeamor.com.br; 3Department of Clinical Oncology, Barretos Cancer Hospital, Barretos 14784400, Brazil; 4Barretos School of Health Sciences Dr. Paulo Prata—FACISB, Barretos 14785002, Brazil; 5Barretos Children’s Cancer Hospital from Hospital de Amor, Barretos 14784400, Brazil

**Keywords:** testicular cancer, germ cell tumors, genomics, epigenomics, pediatric and adult

## Abstract

**Simple Summary:**

Although testicular germ cell tumors (TGCTs) are rare pediatric malignancies, they are the most common malignancies in young adult men. The similarities and differences between TGCTs in adults and children, taking into account the clinic presentation, biology, and molecular changes, are underexplored. In this paper, we aim to provide an overview of the molecular aspects of TGCTs, drawing a parallel between the findings in adult and pediatric groups.

**Abstract:**

Cancer is a leading cause of death by disease in children and the second most prevalent of all causes in adults. Testicular germ cell tumors (TGCTs) make up 0.5% of pediatric malignancies, 14% of adolescent malignancies, and are the most common of malignancies in young adult men. Although the biology and clinical presentation of adult TGCTs share a significant overlap with those of the pediatric group, molecular evidence suggests that TGCTs in young children likely represent a distinct group compared to older adolescents and adults. The rarity of this cancer among pediatric ages is consistent with our current understanding, and few studies have analyzed and compared the molecular basis in childhood and adult cancers. Here, we review the major similarities and differences in cancer genetics, cytogenetics, epigenetics, and chemotherapy resistance between pediatric and adult TGCTs. Understanding the biological and molecular processes underlying TGCTs may help improve patient outcomes, and fuel further investigation and clinical research in childhood and adult TGCTs.

## 1. Introduction

Testicular germ cell tumors (TGCTs) are a distinctive set of diseases in oncology practice due to their curability or the mixture of histologies that appears to reflect embryogenesis. TGCTs are the most common solid tumor in young adults, representing 0.4% of new cases from all sites [1]. In early ages, germ cell tumors represent 3.5% of childhood cancers, occurring in a bimodal distribution with one peak in the first four years of life and a second in adolescence [2,3], and TGCTs represents about 20% of all cases [4]. The incidence rate of testicular germ cell tumors starts to increase in the late teens (10 years old) and reaches its peak in the young adult age group [5,6,7].

TGCTs are classified according to genotype, phenotype, origin cell, and germ cell neoplasia in situ (GCNIS) relationship into three groups. Type I is rare in postpubertal testis and presents as a yolk sac tumor in children less than 6 years old and no precursor cell is identified. Type II is common in postpubertal men in the third and fourth decades of life, with GCNIS as a precursor, which leads to several histologies (see below). Type III usually affects men older than 50 years, and the spermatocitic tumor is a phenotype that is not related to GCNIS. In the current study, we focus on TGCT Types I and II [8].

TGCTs are organized into two main histological groups, known as seminoma (SE) and non-seminoma germ cell tumors (NSGCTs). Seminoma GCTs are made up of undifferentiated germ cells that can histologically resemble sperm and young oogonia, or even germ cells from developmental strains. NSGCTs are subdivided into several histologies, such as embryonal carcinoma, yolk sac tumors (YSTs), teratoma, choriocarcinoma, and mixed NSGCT, in which different histologies are present in different proportions. In contrast to embryonal carcinoma, which histologically resembles the blastocyst, YST has a complex endodermal morphology with embryonic and extraembryonic endodermal components. Mature teratomas are benign tumors, and are the most differentiated, although they may harbor unique neural differentiation as immature teratomas. Finally, choriocarcinoma has trophoblastic differentiation and characteristically features a high level of human chorionic gonadotropin (hCG) in the bloodstream [9].

The clinical presentation of TGCTs usually involves painless swelling of one testis, and is sometimes perceived at a late stage in adults by the partners, and by parents in children. However, it may present with an enlarged tumor or even a palpable abdominal mass when the diagnosis is made at a late stage [10].

Cisplatin is the most important drug used to treat TGCTs and, in recent decades, has changed the natural history of the disease [11,12]. To date, no other drug has outperformed the results of platinum-based combinations; this includes carboplatin, which, in the adult population, has had inferior results [13,14]. Different combinations and doses have been used in protocols for adults and children, often to reduce the acute and late toxic effects, without compromising the outcomes in the latter group [15].

TGCTs have different survival rates according to the age group, with adolescents having a lower rate of event-free survival in a 3 year period (59.9%) compared to children (87.2%) or adults (80.0%) [3,16]. Twenty to forty percent of patients with metastatic testicular germ cell tumors relapse after first-line chemotherapy [11,12]. Furthermore, approximately 50% of these patients can still be cured, and histology, primary tumor location, response to first-line therapy, tumor marker concentrations, and location of metastases (liver, brain, and bone) have been proven to be important prognostic indicator factors in testicular germ cell tumors, in addition to the dose of chemotherapy [17].

Different cytogenetic abnormalities are described when comparing the age of presentation [8]. TGCTs are tumors with a low mutational load, but in their postpubertal presentation, mutations in genes such as *KRAS*, *KIT,* and *TP53* play a role, in addition to the changes in the number of copies of the *KRAS* gene [8,18]. Epigenetics has been the focus of attention in TGCTs. SE and NSGCTs have different methylation patterns, and interest in the role of miRNA is growing, particularly miR-371a-3p and miR-375 as potential biomarkers [18,19,20].

Although most studies have evaluated the biology and molecular changes in adults, and there is a lack of information in children, the molecular mechanisms involved in the development of adult TGCTs are beginning to be defined, and share significant differences and similarities with pediatric GCTs. The search for cytogenetic alterations, epigenetics, and events related to resistance has been explored in TGCTs. In this study, we review the molecular aspects of TGCTs, drawing a parallel between the findings in the adult and pediatric groups.

## 2. Etiopathogenesis of TGCT in Child and Adults

Curiously, the pathogenesis of TGCT begins in utero during embryogenesis, when embryonic stem cells give rise to the primordial germ cells in the genital crest present in the midline of the embryo [21,22]. Pathogenesis differs in some aspects of differentiation, histogenesis, and genomic instability between adults and children. Primordial germ cells are the most implicated in studies on the tumorigenesis of germ cell tumors, and due to their totipotent nature, TGCTs have a wide range of possible histologies. In Type I TGCTs, prepubertal teratomas, as benign tumors, have limited developmental potential and may arise during the migration of primordial germ cells. However, the chromosomal loss of 1p, 4, and 6q, in addition to 1q, 12, 20q, and 22, are implicated in the development of malignant YSTs, the histology most frequently found in testicular tumors in childhood [9,23].

In Type II TGCTs, primordial germ cells and gonocytes may fail to be differentiated in spermatogonia, and the early onset of polyploidization coupled with OCT3/4 switch off failure and changes in genomic imprinting leads to germ cell neoplasia in situ (GCNIS) [24]. The GCNIS pathway is the origin of seminomatous and non-seminomatous TGCTs in adolescents and adults. This pre-invasive precursor of malignant tumors is histologically observed adjacent to normal tissue and is composed of undifferentiated germ cells that proliferate within a seminiferous tubule driven by the testis-specific Y-encoded protein (TSPY) [25,26]. The expression of the KIT ligand by Sertoli cells on the tubules completes the necessary milieu to promote germ cell tumorigenesis [27].

GCNIS commonly progresses to seminomas, and GCNIS and seminomas are able to reprogram the pluripotent embryonal carcinoma, the malignant counterpart of embryonal stem cell [28]. Differentiation steps lead to the emergence of other histologies from embryonal carcinoma, such as postpubertal teratoma, YST, and choriocarcinoma [29,30].

Type III TGCTs are represented by spermatocytic tumors and are more common in patients older than 50 years of age. They arise from differentiated spermatogonia, and the most important event in the oncogenesis appears to be the tumor-specific gain of chromosome 9 and, less frequently, mutations in *HRAS* and *FGFR3* [8,23,31]. Figure 1 shows the etiopathogenesis of the three types of TGCTs.

## 3. Molecular Biology

Because TGCTs are a heterogeneous tumor, it is a challenge to study their genetic basis. However, in recent years, efforts have been made to understand the underlying molecular biology (Table 1) to further improve patient outcomes, particularly for those with chemoresistance and poor risk of disease.

### 3.1. Genetics and Cytogenetics Alterations

Genetic composition of GCTs reflects the embryonic characteristics of the primordial germ cells (PGCs) that give rise to this type of tumor. Although the molecular mechanisms involved in the development of adult TGCTs are beginning to be defined and share significant overlap with pediatric GCTs, little is known about the molecular biology of both, and it is believed that there are important differences that may represent different diseases [9].

TGCTs in adults are characteristically aneuploid, represented by hypertriploids in seminomas and hypotriploids in non-seminomas [32,33]. The most commonly observed change in all histological subtypes of adult TGCTs occurs in the short arm of chromosome 12 (isochromosome 12p), which is the trademark of TGCTs and an important biomarker [8,34,35]. Amplification in the number of copies of the *KRAS* gene reflects frequent alteration of the location of the gene on chromosome 12 and has been described as mutually exclusive with the *KRAS* mutation [36,37].

Other recurring gains at the arm level target chromosomes 7, 8, 21, 22, and X [38,39]. In contrast, in pediatric GCTs (Types I and II), the gain of the short arm of chromosome 12 (i12p) is less frequent, and the most common, but still inconsistent, changes include gains in 1q, 3, 11q, 20q, and 22 [9,40,41,42,43].

The occurrence of loss of heterozygosity (LOH) has been evaluated in TGCTs. In 2010, Vladusic et al. investigated the LOH in suppressor genes, including *CDH1*, *APC*, *TP53*, and *nm23-H1* in TGCT patients (range, 17–60 years). An allelic loss of *TP53* at exon 4 was detected in non-seminomas and LOH of *TP53* at intron 6, *APC*, and *CDH1* was detected in seminomas and non-seminomas. No changes were observed in the *nm23-H1* gene [44]. In the subsequent studies, the same researcher group showed that LOH of the *CDKN2A* was found in two (6%) non-seminomas cases with a yolk sac tumor component, and LOH of the *RB1* was also found in two (6%) non-seminomas with an embryonal carcinoma component [45]. In addition, patterns of LOH were analyzed in TGCT patients (range, 20–52 years) with seminoma components in mixed tumors, pure seminomas, and co-existing GCNIS lesions. LOH in seminoma components in mixed tumors (32%) was more frequent compared to pure seminomas (19%), and the frequencies of LOH at chromosomes 6p and 10q were significantly more frequent in seminoma components in mixed tumors than in pure seminomas [46]. Identification of specific genetic changes, especially events that occurs in adult and pediatric with TGCTs, need to be explored further.

Although TGCTs in adults have a low frequency of recurrent somatic point mutations, the mutational status of pediatric TGCTs is still poorly understood. Unlike most solid tumors, mutations in the *TP53* gene have been rarely described in TGCTs and, when present, they have been associated with a cisplatin-resistant disease, especially in patients with non-seminomatous mediastinal GCTs [47,48,49,50]. Amplifications of the *MDM2* gene have also been associated with mutations in *TP53*, suggesting that mutually exclusive changes in *TP53*/*MDM2* were related to cisplatin-resistant tumors and, consequently, with unfavorable clinical outcomes [51,52]. In contrast, the integrity of the *TP53* gene is considered to be the main reason why GCTs demonstrate high sensitivity to platinum-based chemotherapy [53]. Moreover, other studies have evaluated the presence of mutations in TGCTs, and alterations in *FGFR3*, *AKT1,* and *PIK3CA* genes have been associated with cisplatin-resistant GCTs [54].

*KIT* and *KRAS* mutations are consistently described, suggesting that these are the main conductive mutations of GCTs [18,50,55]. The *KIT* signaling pathway is essential for the normal development and survival of PGCs and spermatogenesis. Mutations in this gene have been reported mainly in seminomas and less frequently in NSGCTs [39,52,56]. In addition, Mata et al. showed that *KIT* mutations in GCTs are associated with *RAS*/*MAPK* pathway driver alterations. This was a retrospective cohort study, in which 568 GCTs patients were analyzed, and 8.1% (46/568) had somatic *KIT* mutations, and the median age at initial pathologic diagnosis of patients with *KIT*-mutant GCTs was 33.4 years (range, 7.2–58.9 years). Among the 46 patients with somatic *KIT* mutations, 23 were men with testicular GCTs, of which six (three seminomas and three NSGCTs) were significantly enriched for oncogenic *RAS*/*MAPK* pathway mutations [57].

The genes of the *RAS* family (*HRAS*, *KRAS,* and *NRAS*) are fundamental for the migration, survival, and progression of the cell cycle, and for activating several downstream routes, including the *RAF*/*MEK*/*ERK* and *PI3K* routes [58,59]. According to the catalogue of somatic mutations in cancer (COSMIC) database, the frequency of *KRAS* and *NRAS* mutations in TGCT is 5% and 3%, respectively. Mutations in the genes of the *RAS* family are more common in seminomas than non-seminomas [36,60,61]. The mutational status of the *BRAF* and *KRAS* was analyzed in 70 pediatric GCT patients (range, 0.5–18 years), of which 17 were testicular; however, no mutations were found [62].

Our group determined the frequency and clinical impact of microsatellite instability status and *BRAF* mutations in 150 TGCT cases, in which the mean age of diagnosis was 30 years (range, 1–63 years). In accordance with the majority of the studies, any case or cell line harboring a *BRAF* mutation was identified, and the microsatellite instability [63]. We also evaluated the presence of the hotspot telomerase reverse transcriptase (*TERT*) gene promoter mutations in more than 130 TGCT cases [64]. We showed for the first time the presence of the *TERT* promoter mutation in four patients (~3%), which is a rare event of TGCTs. Further cohort studies are needed to elucidate these findings and to improve clinical management, leading to better therapeutic alternatives.

Overall, TGCTs in adults and children have a low frequency of somatic mutations and genetic abnormalities is rare. However, adults and pediatric patients showed isochromosome 12p and gain at the chromosome 22 and absence of BRAF mutation. Gains in 1q, 3, 11q, and 20q are frequent in TGCT pediatric patients, whereas gains of 7, 8, 21, and X are frequent in adults. In addition, *KIT*, *RAS* family, *FGFR3*, *AKT1*, *PIK3CA*, *TP53*, and *TERT* mutations have also been demonstrated in adults. Further studies with whole exome sequencing are necessary to identify gene mutations associated with pediatric and adult patients with TGCTs.

### 3.2. DNA Methylation

The epigenetic mechanism in cancer development is well established, with a major focus on DNA methylation. However, compared to other cancers, TGCTs are stated to be hypomethylated, which makes it difficult to uncover methylation-based biomarkers [65,66]. It has been reported that seminomas contain reduced levels of DNA methylation compared to NSGCTs [65,66]. Seminomas show quite low levels of DNA methylation, similar to PGCs or GCNIS, whereas non-seminomas show different extents of DNA methylation. The DNA methylation of NSGCTs has been associated with the degree of differentiation, such as hypermethylation in teratomas, YSTs, and choriocarcinomas, whereas embryonic cell carcinomas show an intermediate pattern [66,67,68]. Smiraglia et al. evaluated the epigenetic differences between seminomas and non-seminomas by restriction landmark genomic scanning and found a level of CpG island methylation in NSGCTs (1.11%) similar to that of other solid tumors, whereas seminomas showed almost no CpG island methylation (0.08%) [66].

To elucidate a set of potential biomarkers in TGCTs, our group evaluated the frequency of methylation of a gene panel (*VGF*, *MGMT*, *ADAMTS1*, *CALCA*, *HOXA9*, *CDKN2B*, *CDO1,* and *NANOG*) in primary TGCT samples, which included seminomas and NSGCTs. We observed a high frequency of *MGMT* and *CALCA* methylation in NSGCTs and demonstrated for the first time that *CALCA* methylation is associated with non-seminoma tumors, refractory disease, and poor clinical outcome in TGCT patients (range, 26–32 years) [69]. Similar to our results were those obtained by Sanjay et al., in which the characteristic promoter hypermethylation signatures in male germ cell tumors were determined via analysis of CpG islands of 21 gene promoters by methylation-specific PCR in seminoma and non-seminoma GCTs. NSGCTs showed 60% of methylation in one or more gene promoters, including *MGMT*, *RASSF1A*, *BRCA1,* and a transcriptional repressor gene *HIC1*. In contrast, seminomatous tumors showed a near-absence of methylation [70]. Promoter hypermethylation of *RASSF1A* and *HIC1* was associated with tumors resistant to cisplatin-based regimens in a cohort of GCTs, whereas *MGMT* and *RARB* were sensitive [71].

Some of the epigenetic markers have been explored as diagnostics of TGCTs. The methylation status of *XIST* was investigated, and unmethylated *XIST* sequences were detected in most tissues (30/31) (seminoma and non-seminoma) and plasma (16/25) samples from patients with TGCTs, contrasting with peripheral blood lymphocytes from non-germ cell tumors of individuals (no detection). The Nodal co-receptor Cripto is another useful serological marker proposed for TGCT diagnosis. Hypomethylation of the *CRIPTO* promoter was found in undifferentiated fetal germ cells, embryonal carcinomas, and seminomas, whereas hypermethylation was associated in differentiated fetal germ cells and the differentiated types of NSGCTs [72]. These results suggest the methylation status of *XIST* and *CRIPTO* could be a tumor marker for detection and monitoring of TGCTs, however, confirmation from a larger series is necessary [73].

Methylation studies involving adults are significantly more common than those of children. In 2003, Kato et al. evaluated promoter methylation in yolk sac tumors from infants and the results showed that methylation of *RUNX3* was detected in 80% of infantile YSTs examined, and no adult GCTs showed *RUNX3* methylation [74]. In 2006, the same research group showed that the *APC* promoter was methylated in 7 (70%) of 10 infantile YSTs. Taken together, these results suggest that *RUNX3* and *APC* are the tumor suppressors involved in the pathogenesis of testicular YSTs in infants (Kato et al., 2006) [75].

The heritability of the global genomic methylation phenotype in families and the association between global (*LINE-1*) methylation levels and testicular cancer was examined [76]. The heritability of *LINE-1* methylation may be gender-specific and there was a marginally significant inverse association between *LINE-1* hypomethylation levels and increased TGCT risk [76].

The differences in methylation between the histologic subtypes of pediatric GCTs were analyzed using in an epigenome-wide study. The differentially methylated regions (DMRs) were identified in a set of 154 pediatric tumors, including germinomas/seminomas/dysgerminomas, teratomas, YSTs, and a mixed histologic subtype from gonadal, extragonadal, and intracranial locations. A total of 8481 DMRs were identified (family-wise error rates—FWER < 0.05) and distinct differences in gene-specific methylation were found between the histologic subtypes of GCT, particularly between germinoma/seminoma/dysgerminoma and YST. Pathway analysis on the top 10% of genes with differential methylation suggested angiogenesis and immune cell-related pathways, which displayed decreased methylation in germinomas/seminomas/dysgerminomas relative to YST. Therefore, the genes that are differentially methylated may provide insights into GCT etiology [77].

In summary, most works on altered DNA methylation patterns in TGCTs have been performed in adults showing methylation in the following genes: *VGF*, *MGMT*, *ADAMTS1*, *CALCA*, *HOXA9*, *CDKN2B*, *CDO1, NANOG, RASSF1A*, *BRCA1, HIC1, RARB, XIST,* and *CRIPTO*. Few studies have been realized in TGCT pediatric patients, and only *RUNX3* and *APC* methylation has been described. There is a wide information gap between TGCTs in adult and children, and only one gene (*LINE-1*) was reported to be hypomethylated in both age groups. Overall, the elucidation of epigenetic alterations in TGCTs including the study of mutations could improve the usefulness of methylation-based biomarkers.

### 3.3. miRNAs as Tumor Biomarkers

Recent studies have suggested an involvement of microRNAs (miRNAs) as tumor biomarkers and therapeutic targets. MiRNAs comprise a group of small (approximately 22 nucleotides) non-coding RNAs that regulate the expression of protein-coding genes by binding to their target messenger RNAs (mRNAs), thus resulting in translational repression or mRNA degradation [78,79]. Several studies have revealed the role of miRNAs in a variety of biological processes, including development, differentiation, viral infections, and cancer [80]. There are more than 1000 mature miRNAs [81] and some of them have been associated with malignant-GCTs as a new generation of biomarkers [19].

In 2006, Voorhoeve et al. performed a screen for miRNAs that cooperate with oncogenes in cellular transformation, and found miR-372 and miR-373 were particularly abundant in GCT tissue and cell lines [82]. In agreement with these results, Palmer et al. determined the global miRNA profiles in pediatric GCTs arising at both gonadal and extragonadal sites, and compared the changes observed with those reported for adult gonadal malignant GCTs [83]. The most significant differentially expressed miRNAs in malignant GCTs were all from the miR-371~373 and miR-302 clusters, which were over-expressed regardless of histological subtype (yolk sac tumor/seminoma/embryonal carcinoma), site (gonadal/extragonadal), or patient age (pediatric/adult). Subsequently, the potential of miR-371~373 and miR-302/367 as biomarkers of malignant GCTs was reported [84]. In a multicentric study, serum samples of 616 patients with TGCTs (range, 16–69 years) and 258 male controls were examined for serum levels of miRNA-371a-3p. The results showed that the measurement of serum miR-371a-3p levels provides both a sensitivity and a specificity greater than 90% and an area under the curve (AUC) of 0.96 [20]. These data support the use of miRNAs as tumor biomarkers.

The use of circulating miR-371a-3p was analyzed as a marker for malignant germ cell tumor management in TGCT patients (range, 26–40 years) prior to orchiectomy [85]. Performance characteristics of serum miR-371a-3p were compared with predictable markers, including α-fetoprotein, β-human chorionic gonadotropin, and lactate dehydrogenase. The miR-371a-3p test showed a specificity of 100%, sensitivity of 93%, and AUC of 0.978 [85]. These results confirm the previously published studies and suggest the effectiveness of a positive miR-371a-3p test post-chemotherapy.

Micro-RNA expression was also associated with cisplatin-resistant germ cell tumor cell lines. There is an association of the upregulation of miR-512-3p/-515/-517/-518/-525 and down-regulation of miR-99a/-100/-145 with a cisplatin-resistant phenotype in human germ cell tumors [86]. However, it will be interesting to evaluate tumor samples of patients with germ cell tumors with both cisplatin-sensitivity and resistance to analyze whether all of these miRNAs are also found in vivo.

Similarly, studies have paid particular attention to miR-371~373 and miR-302 clusters across age groups (pediatric and adult patients). Further experimental support is required to confirm the high sensitivity and specificity of miRNAs as biomarkers of TGCTs and, with this confirmation, the miRNAs will be considered clinical biomarkers to overcome the uncertainty of equivocal scenarios, for which the rate of uncertainty is concerning. Finally, miRNAs may improve the quality of the care of patients, contributing to personalized and precise medicine.

### 3.4. Molecular Implications Responsible for Chemotherapy Sensitivity and Resistance in TGCTs

Due to the efficacy of cisplatin treatments for TGCTs and the high five-year survival rate, TGCTs are considered curable neoplasms [89]. However, resistance to chemotherapeutic treatment still appears in around 20% of patients with metastatic disease [90].

Studies have focused on the molecular events responsible for the cisplatin (CDDP) resistance mechanisms, in addition to the high chemotherapy sensitivity of TGCTs, to develop a more effective treatment for patients with metastatic cancers. Based on this statement, we pose the following question: Why are some TGCT patients so sensitive to CDDP and resistant to others?

The resistance of TGCT to chemotherapy has been related to different molecular mechanisms, including mutations, karyotype abnormalities, and epigenetic aberrations. Cisplatin acts via covalent binding to the DNA molecule, which is recognized by proteins participating in the process of DNA repair, leading to cell cycle arrest and apoptosis [91].

The sensitivity to CDDP might be explained by the inherent properties of the embryonal stem cells (ESCs) [92], which react to DNA damage through the elimination of damaged cells by apoptosis [93,94]. Therefore, the loss of the embryonic properties might underlie the development of treatment resistance.

Studies have shown a model of cisplatin-sensitive GCT. When the cisplatin-sensitive germ cell tumor cells suffer DNA damage (e.g., cisplatin), the DNA damage response system is activated, in which *TP53* is activated and a mediated response occurs via induction of *PUMA* and *NOXA*, leading to apoptosis. The apoptosis response is enhanced by the presence of transcription factor *OCT4,* and high levels of *OCT4* are associated with low expression of *P21* and failure to activate a G1 arrest [91]. In contrast to other solid tumors, GCT is rarely characterized by *TP53* mutation. One study showed that in relapsed GCTs, *TP53* mutations were detected in about 14% of tumors (4 of 28 tumors), for which three were mature teratomas and the other was a secondary non-germ-cell malignancy derived from a teratoma [48]. Kersemaekers et al. investigated the role of *TP53* and *MDM2* in the treatment response of patients (range, 17–56 years) with germ cell tumors. Mutation analysis revealed only one silent *TP53* mutation in one of the responding patients. All embryonal carcinomas were homogeneously positive for *MDM2*, whereas the other histologic components were heterogeneous. Only one patient (1/12) with embryonal carcinomas showed *MDM2* amplification. Although the presence of wild-type *TP53* was detected in TGCTs, there is no correlation between the high level of *TP53* and treatment sensitivity, and a *TP53* inactivation is a rare event in the development of cisplatin resistance [51].

Studies have suggested that mutations in *TP53* and overexpression of *MDM2* may happen in proportion to cisplatin refractory TGCTs; however, the extent of these mutations is not clear [50,52]. Therefore, the role of *TP53* mutations in cisplatin resistance of TGCTs has been controversial and more studies are required.

Defective mismatch repair (MMR), microsatellite instability, and *BRAF* mutations have been associated with relapse and treatment failure in TGCT patients (range, 18–55 years), who show a decrease in the MMR gene expression (*MLH1*, *MLH2*, or *MSH6*) [95].

Feldman et al. attempted to validate the frequency of *BRAF* mutations among GCT patients. Adolescent and adult patients (range, 14–60 years) with GCT who received cisplatin-based chemotherapy were classified as cisplatin-sensitive and cisplatin-resistant. Nineteen mutations were detected in GCTs patients, but no *BRAF* mutations were identified. Somatic mutations within *KRAS*, *AKT1*, *PIK3CA*, and *HRAS* were exclusive to cisplatin-resistant patients [54]. Genomic evolution and chemoresistance was analyzed by Taylor-Weiner et al. using whole-exome and transcriptome sequencing of precursor, primary (testicular and mediastinal), and chemoresistant metastatic human GCTs (n = 49 patients; range, 17–57 years) [50]. Mutational significance analysis showed *KRAS* as the most significant altered gene and primary TGCTs were uniformly wild type for *TP53*. Moreover, TGCTs with chemotherapy resistance showed an additional reciprocal loss of heterozygosity, which was associated with loss of *NANOG* and *POU5F1* markers [96,97] in chemoresistant teratomas or transformed carcinomas. These results suggest that different genomic features underlying the origins of GCTs are related with the chemosensitivity phenotype [50].

Accumulating evidence suggests that cisplatin resistance is associated with an increase in DNA methylation. The promoter hypermethylation of *RASSF1A* and *HIC1* genes have been related to resistance of GCT, whereas the transcriptional inactivation of *MGMT* confers sensitivity to cisplatin [71].

The mechanisms of cisplatin hypersensitivity and resistance in embryonal carcinoma were explored in human testicular cancer-derived EC cell lines. The data shows that repression of *H3K27* methylation is a mechanism of cisplatin acquired resistance in TGCTs and that restoration of *PRC2* function is a potential alternative to overcome treatment failure [98].

Cisplatin resistance is most likely multifactorial and is a challenge in the clinical approach to TGCT patients, especially in pediatric patients for whom there is a lack of research. There is a scarcity of large molecular analysis focusing on cisplatin resistance because targeting a single marker is not sufficient to reverse the phenotype. A better understanding of cisplatin resistance could improve the testing of new drugs and targeted therapies with better clinical benefit.

## 4. In Vitro and In Vivo Models

In recent years, major technological advancements have led to a better understanding of the molecular mechanisms of TGCTs. However, this progress has had a slight impact on the cancer therapeutic approach, probably due to the limitation of experimental models to predict efficacy in clinical trials. In an effort to offset this limitation, the interest in the development of different TGCT models is increasing.

It remains a challenge for the clinic to investigate the molecular and genetic mechanisms involved in the development of cisplatin resistance in TGCTs, for which the frequency of recurrence is low and the availability of histological samples post-chemotherapy is scarce, because surgical resection of the tumor is the first line of treatment [99]. In general, cisplatin resistance in TGCTs is commonly studied in primary tumors of patients who may develop them at some point in the future, which is not ideal due to its naive relationship with chemotherapy [100].

Other strategies to investigate the mechanisms associated with the resistance acquired by tumor cells to cisplatin include the use of preclinical models, in vitro and in vivo, obtained from the cultivation and exposure of TGCT cell lines to incremental doses of the drug, for long periods of time [101], in addition to the use of animal models that reproduce the phenotypic properties of the human tumor [102]. Although in vitro cell culture systems have been used extensively for decades, they represent oversimplified models, which are characterized by the absence of heterogeneity and lack of microenvironment components [103]. Therefore, it is crucial to develop more accurate and clinically relevant mice models that genuinely represent TGCTs in adult and pediatric patients, according to their etiopathogenesis, histopathology, and metastatic progression, and the response of therapy. In this context, patient-derived xenograft (PDX) models have been used as an outstanding alternative [104,105]. PDX model development is generated via the transplant of primary tumor fragments or tumor-derived cancer cells from the patient into immunocompromised mice [106,107], and has been established in different types of tumors. In TGCTs, PDX models have been developed with a focus on mouse models of resistant disease, which may be established by injecting the cisplatin-resistant clones of TGCT cell lines or by implanting cisplatin-resistant human tumors [102,108,109].

Different models of chemoresistant TGCT cell lines have already been developed and studied for genotypic and phenotypic changes [99]. NTERA-2 and NCCIT cisplatin-resistant cell lines were injected into immunodeficient mice and disulfiram was used to examine chemosensitization of resistant cell lines. Disulfiram in combination with cisplatin showed synergy for NTERA-2 and NCCIT cisplatin-resistant cells and inhibited the growth of NTERA-2 (cisplatin-resistant) xenografts. High *ALDH1A3* expression and increased *ALDH* activity were detected in both refractory cell lines. In addition, the upregulation of the *ALDH* isoform *ALDH1A3* was confirmed in 216 patient samples with all histological subtypes of testicular tumors. These results suggest *ALDH1A3* as a novel therapeutic treatment in TGCTs, and disulfiram represents a feasible treatment option for refractory TGCTs [109].

Our group developed an in vitro model of cisplatin resistance to identify new potential therapies for TGCT-resistant patients (data not published). We established a CDDP-resistance model using the NTERA-2 cell line (NTERA-2R), which was treated for approximately eight months with incremental doses of CDDP. We then performed a phenotypical characterization and NTERA-2R exhibited a significant increase in cell proliferation capacity, augmented clonogenic survival, and higher migration ability, suggesting an aggressive phenotype. To elucidate the molecular changes associated with CDDP-resistance, we analyzed the expression of genes related to damage and repair mechanisms. Compared to the parental cell line, NTERA-2R showed several differentially expressed genes related to DNA repair and cell cycle regulation. These results support the idea that the main change in NTERA-2R is possibly an increased DNA repair capacity and specific changes in cell cycle control, which may trigger apoptosis evasion and allow cells to proliferate, even in the presence of CDDP adducts.

Changes in the cell cycle (increase in G1 and decrease in the S phase), increase in the number of acquired mutations (mainly in the *ATRX* gene), changes in the gene expression pattern, and chromosomal variation (gain of 12p, 1, 17, 20, and 21 loss of X) were also observed in the resistant NCCIT strain [100].

To investigate cisplatin-resistance genetic basis in TGCT, Piulats et al. implanted a collection of matched cisplatin-sensitive and -resistant non-seminoma tumors in nude mice and compared the genomic hybridization (CGH). Comparative CGH analyses showed a gain at the 9q32-q33.1 region, and the presence of this chromosomal rearrangement was correlated with poorer overall survival (OS) in metastatic germ cell tumors. Moreover, *POLE3* and *AKNA* genes were deregulated in resistant tumors harboring the 9q32-q33.1 gain. Therefore, the cisplatin-refractory orthoxenografts of TGCTs are potent models to test the efficiency of drugs, and identify prognosis markers and gene alterations [102].

An immunohistochemical study investigated xenograft models and NSGCT samples with a focus on OCT4-negative cells with undifferentiated EC morphology and their association with chemotherapy resistance [108]. Subcutaneous xenograft tumors of the NSGCT cell lines H12.1 (cisplatin-sensitive) and 1411HP (cisplatin-resistant) were established in athymic nude mice. The cisplatin-sensitive cell line H12.1 leads to xenografts in which EC structures are mainly composed of OCT4-positive cells, whereas xenografts from the resistant cell line 1411HP exclusively comprise OCT4-negative EC areas, suggesting that the growth of NSGCTs in patients with cisplatin-refractory disease may be determined by OCT4-negative EC cells [108]. In addition to this data, a mouse TGCT model featuring germ cell-specific Kras activation and Pten inactivation was developed as a representative model of malignant TGCTs in men. The resulting mice developed malignant, metastatic TGCTs composed of teratomas and embryonal carcinomas, the latter of which exhibited stem cell characteristics, including expression of the pluripotency factor OCT4 [110].

The need for new therapeutic options for patients with natural or acquired resistance to cisplatin has led to the investigation of the activity of different compounds (kinase inhibitors directed at mTOR, EGFR, HER2, VEGFR, and IGF-1R), in sensitive (H12.1 and GCT72), and resistant (H12.1RA, H12.1D, 1411HP, and 1777NRpmet) cell lines of TGCT. Research has shown that these compounds have potential activity when used alone, but not when in combination with cisplatin [111].

Despite these recent advances in the use of mouse models to study TGCTs, such models must be developed for pediatric patients, and new molecular studies must be performed to provide powerful experimental tools to prioritize new therapeutic approaches for future clinical trials. Figure 2 summarizes the comparison of clinical and molecular differences between adult and pediatric patients with TGCTs as hallmarks of cancer.

## 5. Conclusions

Clinic presentation of TGCTs is highly similar between children and adults, although their identification and search for elucidation could be driven by different contexts. GCNIS is the main player in adult tumorigenesis, in contrast to child tumors, which arise from non-GCNIS pathways. The relative paucity of molecular studies in the pediatric group poses an inferential limitation to compare children and adults. However, advances in recent years provide a new perspective. Thus, miRNAs and methylation can be identified as similarities to highlight that are deserving of more attention.

Considering genetic abnormalities, TGCTs in adults and children have a low frequency of somatic mutations, and genetic abnormalities are rare. However, adults and pediatric patients showed isochromosome 12p and gain at chromosome 22, and absence of *BRAF* mutation. Studies of adults have shown mutation in *KIT*, *RAS* family, *FGFR3*, *AKT1*, *PIK3CA*, *TP53*, and *TERT*. In contrast, there is a scarcity of research in TGCT pediatric fields.

Several epigenetic alterations have been demonstrated to impact TGCTs, and DNA methylation and microRNAs have been the most frequent targets of studies. Promoter methylation of *VGF*, *MGMT*, *ADAMTS1*, *CALCA*, *HOXA9*, *CDKN2B*, *CDO1*, *NANOG*, *RASSF1A*, *BRCA1*, *HIC1*, *RARB*, *XIST*, and *CRIPTO* have been reported in adults. Again, there is a lack of information in TGCT pediatric fields, and only *RUNX3* and *APC* methylation have been described. Moreover, miRNAs, such as miR-371a-3p, have been shown to be expressed and detectable in the blood of adult and pediatric patients with viable GCTs. Thus, deep knowledge of the epigenetic mechanisms underlying the development of TGCTs may lead to new therapeutic approaches.

Knowledge of the biological and molecular insights underlying TGCTs may help improve patient outcomes and may fuel further translational and clinical research in childhood and adult TGCTs. Due to the rarity of TGCTs, it is hard to establish a large cohort of adults and, in particular, children, to compare the groups. However, collaborative efforts should be made to assemble these groups to enable a better etiopathogenic understanding involving children, adolescents, and adults. In addition, new therapeutic approaches might be achieved with the development of child in vivo models, thus leading to more effective and less toxic treatment protocols.

## Figures and Tables

**Figure 1 cancers-13-02349-f001:**
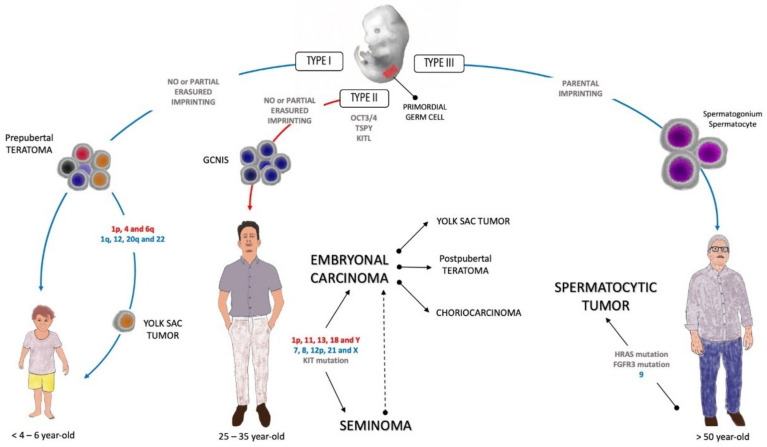
Etiopathogenesis of testicular germ cell tumors (TGCTs). The red letters mean chromosomic loss, and the blue letters chromosomic gain. The blue arrows represent the non-GCNIS pathway and the red arrow represents the GCNIS pathway. GCNIS: germ cell neoplasia in situ; KITL: KIT ligand; TSPY: testis-specific Y-encoded protein.

**Figure 2 cancers-13-02349-f002:**
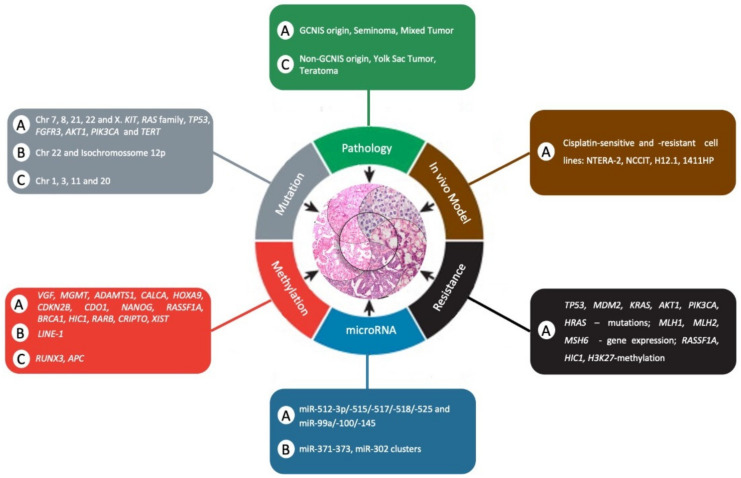
Comparison of clinical and molecular differences between adult and pediatric patients with TGCTs as a hallmark of cancer. The letter “A” represents adults, “C” represents child, and “B” represents both adult and child. Adapted from Hanahan and Weinberg [112].

**Table 1 cancers-13-02349-t001:** Epigenetic-based biomarkers in testicular germ cell tumors in adult and pediatric patients.

Adult	Pediatric
Biomarker	Major Findings	Biomarker	Major Findings
DNA Methylation			
*VGF, MGMT, ADAMTS1,**CALCA, HOXA9,**CDKN2B, CDO1,* and *NANOG*	Hypermethylation of *MGMT* and *CALCA* promoters associateswith non-seminoma and poor prognosis *CALCA* associateswith refractory disease. [69]	*RUNX3*	*RUNX3* promoterhypermethylation was detected in YST in infants (80%). [74]
*MGMT*, *RASSF1A*, *BRCA1,* and a transcriptional repressor gene *HIC1*	Non-seminoma showed methylation in *MGMT*, *RASSF1A*, and *BRCA1* and *HIC1*. Seminoma showed a near-absence of methylation. [70]	*APC*	*APC* promoter hypermethylation was detected in YST in infants (70%). [75]
*RASSF1A*, *HIC1*, *MGMT,* and *RARB*	Hypermethylation of *RASSF1A* and *HIC1* was associated with tumors resistant to cisplatin-based regimens, whereas *MGMT* and *RARB* were sensitive. [71]	Epigenome-wide study	DMRs were identified in a set of 154 pediatric tumors from gonadal, extragonadal and intracranial locations. [77]
*XIST*	Unmethylated DNA *XIST* fragments in seminoma and non-seminoma. [73]		
*CRIPTO*	Hypomethylation in undifferentiated fetal germ cells, embryonal carcinoma and seminomas. Hypermethylation in differentiated fetal germ cells and the differentiated types of non-seminomas. [72]		
*LINE-1*	Strong correlation in *LINE-1* methylation levels among affected father-affected son pairs. *LINE-1* hypomethylation was associated with the risk of testicular cancer. [76]
**Genetic abnormalities**			
Isochromosome 12p	The most commonly observed change in all histological subtypes of TGCTs. [8,34,35]	Isochromosome 12p	Less frequent in types I and II. [40,41,42].
Chr 7, 8, 21, 22, and X	Gains at the arm level target. [38,39]	Chr 1, 3, 11, 20, and 22	Gains in 1q, 3, 11q, 20q, and 22 are common, but still inconsistent. [9,43]
*RAS* family (*HRAS*, *KRAS,* and *NRAS*)	More common in seminoma when compared to non-seminoma [36,60,61].		
*TP53*	Rarely described in GCTs but, when present, they were associated with a cisplatin-resistant disease, especially in patients with non-seminoma mediastinal [47,48,49,50].		
*FGFR3*, *AKT1*, *PIK3CA*	Associated with cisplatin-resistant GCTs [54].		
*TERT*	*TERT* promoter mutation is rare [64].		
*KIT* and *KRAS*	*KIT* mutations in GCTs are associated with *RAS*/*MAPK* pathway driver alterations [57].
*BRAF*	*BRAF* mutation was absent [62,63,87,88].
**microRNA**			
miR-372 and miR-373	miR-372 and miR-373 were particularly abundant in GCT tissue and cell lines. [82]	miR-371~373 and miR-302 clusters	miR-371~373 and miR-302 clusters were overexpressed regardless of histological subtype, site (gonadal/extragonadal), or patient age (pediatric/adult) [83].
miR-371~373 and miR-302/367	miR-371~373 and miR-302/367 as biomarkers of malignant GCTs were reported [84].		
miR-371a-3p	Serum miR-371a-3p levels provide both a sensitivity and a specificity greater than 90% and an area under the curve (AUC) of 0.96 [20].The miR-371a-3p test showed a specificity of 100%, sensitivity of 93%, and AUC of 0.978 [85].

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
