# Peer review of "Molecular Biology of Pediatric and Adult Male Germ Cell Tumors"

_cancers, 2021, doi:10.3390/cancers13102349_

Round 1

Reviewer 1 Report

This is a review paper focused on the biology of pediatric and adult testicular germ cell tumors, comparing these two types of TGCT in between.

At the most beginning, it should be clarified which tumors the authors really address. In childhood, testicular tumors represent only a small proportion of germ cell tumors in children and it is often difficult to get separated data for them. Mature teratomas are benign tumors and must be distinguished from yolk sack tumors, as the most common testicular malignant tumors in children. In adults, spermatocytic seminoma (TGCT type III) differ from the rest of TGCT in adults (TGCT type II) and cannot be really put together. A clear overview of GCT classification should be given at the beginning of the introduction.

The review brings a large overview of results of studies published on TGCT biology and cisplatin resistance and may act as a good source of reference. Unfortunately, it is not quite well structured, attempts on some kind of synthesis of the results and drawing of general conclusions are mostly missing. The main aim of the review – to depict the parallels and differences between childhood and adult TGCT – is often hard to trace throughout the review. The review would definitely benefit from re-writing and better delineation of the idea and results.

Particular points:

In the first paragraph of introduction devoted to the incidence of TGCT, at least some values on overall incidence should be added.

In the next paragraph on TGCT histology, the explanation of NS histology subtypes is not well organized and addressed, would benefit of re-writing.

In Figure 1, YST does not developed from prepubertal teratoma in children, as it implies.

In Molecular Biology part – TGCT are not only hypertriploid but also hypotriploid (NS).

The information given of Figure 2 is a bit chaotic, particularly in the clinicopahological part.

The English needs some editing and re-writing. There is no reason to write the histological subtypes of TGCT with capital letters (at the beginning of the paper).

Author Response

1) At the most beginning, it should be clarified which tumors the authors really address. In childhood, testicular tumors represent only a small proportion of germ cell tumors in children and it is often difficult to get separated data for them. Mature teratomas are benign tumors and must be distinguished from yolk sack tumors, as the most common testicular malignant tumors in children. In adults, spermatocytic seminoma (TGCT type III) differ from the rest of TGCT in adults (TGCT type II) and cannot be really put together. A clear overview of GCT classification should be given at the beginning of the introduction.

Response: Thank you for pointing that out. We have kindly been invited by Cancers to write about testicular cancer and in line of our experience putting adult and children oncologist to think together in clinics and research, we aimed focus on testicular cancer during the manuscript. We are in agreement that is often difficult to get separated data for testicular germ cell tumors in children and there is a limitation for conclusions for sure. However, we can see a great opportunity to drive some hypothesis generation from our review. You can find along our manuscript the term Testicular germ cell tumors (TGCTs). Some modifications during the manuscript have been made to suppress any other reference about non testicular germ cell tumors. Therefore, we believe it were even more clear which tumor we address.

In the page 3, topic 2, we rephrase to highlight the different nature of mature teratoma and yolk sac tumors.

We have discussed the GCTs classification in the topic "Etiopathogenesis of TGCT in child and adults". In addition, we are in agreement that is pertinent it should be given at the beginning of the introduction.

2) The review brings a large overview of results of studies published on TGCT biology and cisplatin resistance and may act as a good source of reference. Unfortunately, it is not quite well structured, attempts on some kind of synthesis of the results and drawing of general conclusions are mostly missing. The main aim of the review – to depict the parallels and differences between childhood and adult TGCT – is often hard to trace throughout the review. The review would definitely benefit from re-writing and better delineation of the idea and results.

Response: Thank you for pointing that out. We have included in the end of each topic one paragraph with the most relevant message about pediatric and adult TGCT. In addition, we also performed a synthesis of the results in the general conclusion. 

3) In the first paragraph of introduction devoted to the incidence of TGCT, at least some values on overall incidence should be added.

Response: We have included the incidence of TGCTs in the first paragraph of introduction.

4) In the next paragraph on TGCT histology, the explanation of NS histology subtypes is not well organized and addressed, would benefit of re-writing.

Response: We have rewritten the explanation of NS histology to make it even clearer.

5) In Figure 1, YST does not developed from prepubertal teratoma in children, as it implies.

Response: Thank you for your assertion and point of view. In fact, the data to support this statement is scarce in the literature, manly due the lack of in vivo models in Type I germ cell tumor studies and the rarity of these tumors.

It is worth mentioning that in the publication of Pierce et al. (2016) (Reference 9 in the manuscript) they showed at Figure 1 the possibility of YST develop from teratomas components. In addition, the last "WHO Classification of Tumours of the Urinary System and Male Genital Organs" edited by Holger Moch et al. in name of World Health Organization, refers the same development of YST, which is shown in the Figure 4.04, page 191. Literally, the description of Figure 4.04 is as following: "The yolk sac tumours unrelated to GCNIS also show specific chromossomal gains and losses, and result from progression of the teratoma component". Therefore, we can conclude that until this moment, the scarce evidences point out to YST could originate from teratoma components. To make all of this information clearer, we have changed Figure 1.

6) In Molecular Biology part – TGCT are not only hypertriploid but also hypotriploid (NS).

Response: We totally agree with the reviewer and the sentence has been restructured - “TGCTs in adults are characteristically aneuploid, represented by hypertriploid in seminoma and hypotriploid in nonseminomas”.

7) The information given of Figure 2 is a bit chaotic, particularly in the clinic pahological part.

Response: We have performed another figure layout and the clinic pathological part have been changed. Therefore, we would like the reviewer to consider our efforts to do a relevant figure according to hallmarks of cancer.

8) The English needs some editing and re-writing. There is no reason to write the histological subtypes of TGCT with capital letters (at the beginning of the paper).

Response: Our manuscript was edited and rewritten by MDPI English editing service (English editing ID: English-28079).

Reviewer 2 Report

General points:

I would suggest to subdivide the sections in ‘pediatric GCT’ and ‘adult GCT’ and show the differences between them. As the authors also talk about mediastinal GCT it might be worthwhile changing the title into ‘Molecular biology of pediatric and adult male germ cell tumors’.

Major Points:

Introduction

  • The introduction needs to be shortened: The authors should focus on what they want to show. What’s the main topic of the review? Is it the difference in the molecular biology between pediatric and adult GCTs?
  • Line 51: Please list all subtypes of NSGCT: yolk sac tumors, embryonal carcinoma, teratoma, choriocarcinoma and mixed NSGCT.
  • Line 76 - 82: for the adult survival rates and prognostic criteria please refer to the IGCCCG classification (International Germ Cell Cancer Collaborative Group).

Molecular biology

  • It is often unclear if the authors talk about pediatric or adult GCT. An idea would be to subdivide the topics into ‘pediatric’ and ‘adult’ GCT
  • Line 165: when referring to cisplatin resistance please also cite ‘Genomic evolution and chemoresistance in germ cell tumours’ by Taylor-Weiner et al, Nature 2016
  • What about loss of heterozygosity
  • Line 186: was there a difference between adult and pediatric GCT with respect to frequency?
  • Line 45: It is the most common solid tumor in young adults and in early ages, occurring in a bimodial distribution with……..
  • Line 53: ‘YolkSac tumor’: Please use abbreviation YST
  • Line 64: Please explain SRY
  • Line 89: please write ‘miR371a-3p instead of ‘miR 371a’
  • Line 133: please delete ‘the’ -->due to TGCT being a heterogeneous tumor
  • Line 133: please delete ‘the’ -->genetic composition of (the) GCTs..
  • Line 136: please write: mainly for those with chemoresistant and poor risk disease
  • Line 159: mediastinal GCTs are not testicular GCT, so please delete the ‘T’ --> mediastinal GCT
  • Table 1 is good but gain of chromosome 12p is not a mutation

Author Response

1) I would suggest to subdivide the sections in ‘pediatric GCT’ and ‘adult GCT’ and show the differences between them. As the authors also talk about mediastinal GCT it might be worthwhile changing the title into ‘Molecular biology of pediatric and adult male germ cell tumors’.

Response: We appreciate the suggestion of the reviewer to change the title of our manuscript and the alteration was performed. We respect the reviewer’s opinion about to subdivide the manuscript in pediatric and adult section, however, we believe that the manuscript would be inconsistent and out of proportion to across age groups (pediatric and adult patients). We are been convinced by the number of published studies and information in TGCT pediatric is really lower compared to TGCT adults. Therefore, the ‘pediatric section’ would be short and with few details.

To highlight the differences between pediatric and adult patients with TGCTs, we have included in the end of each topic one paragraph with the most relevant message about pediatric and adult TGCT.

2) The introduction needs to be shortened: The authors should focus on what they want to show. What’s the main topic of the review? Is it the difference in the molecular biology between pediatric and adult GCTs?

Response: We appreciate the reviewer's suggestion and the introduction has been shortened according to the focus of our review. The main topic of our review is highlighting the major similarities and differences in cancer genetics, cytogenetics, epigenetics, and chemotherapy resistance, between pediatric and adult TGCTs.

3) Line 51: Please list all subtypes of NSGCT: yolk sac tumors, embryonal carcinoma, teratoma, choriocarcinoma and mixed NSGCT.

Response: We have rewritten the explanation about NS histology to make it even clearer and all subtypes of NSGCT were listed.

4) Line 76 - 82: for the adult survival rates and prognostic criteria please refer to the IGCCCG classification (International Germ Cell Cancer Collaborative Group).

 Response: Thank you for pointing that out. We have referenced IGCCCG classification.

5) It is often unclear if the authors talk about pediatric or adult GCT. An idea would be to subdivide the topics into ‘pediatric’ and ‘adult’ GCT

Response: We are grateful for this suggestion and this issue was addressed in comment 1.

6) Line 165: when referring to cisplatin resistance please also cite ‘Genomic evolution and chemoresistance in germ cell tumours’ by Taylor-Weiner et al, Nature 2016

Response: We agree with the reviewer suggestion and the Taylor-Weiner et al., 2016 study has been referenced.

7) What about loss of heterozygosity

Response: Thank you for pointing that out. Studies with loss of heterozygosity (LOH) were missing in our review and in this new version, the main studies in TGCT were added in the topic “Genetics and Cytogenetics alterations”.

8) Line 186: was there a difference between adult and pediatric GCT with respect to frequency?

Response: In our study we analyzed 150 TGCT cases (range, 1-63 years) and any of them harboring a BRAF mutation, as well as the microsatellite instability in adult and pediatric patients. Moreover, when we evaluated the presence of the hotspot telomerase reverse transcriptase (TERT) gene promoter mutations in TGCT cases and all them were adults. TERT promoter mutation was found only in four adult patients (~3 %). Therefore, we do not have the frequency of TERT in pediatric patients. The paragraph with our studies was changed.

9) Line 45: It is the most common solid tumor in young adults and in early ages, occurring in a bimodial distribution with……..

Response: The sentence has been corrected.

10) Line 53: ‘YolkSac tumor’: Please use abbreviation YST.

Response: The abbreviation YST has been used along the text once it was defined for the first time in the Introduction section.

11) Line 64: Please explain SRY.

Response: Due to the fact that our Introduction has been shortened according to the reviewer’s suggestion, the paragraph containing the abbreviation “SRY” has been removed.

12) Line 89: please write ‘miR371a-3p instead of ‘miR 371a’

Response: The miR 371a has been replaced by miR-371a-3p.

13) Line 133: please delete ‘the’ -->due to TGCT being a heterogeneous tumor

Response: Grammatical article ‘the’ has been deleted in the text.

14) Line 133: please delete ‘the’ -->genetic composition of (the) GCTs.

Response: Grammatical article ‘the’ has been deleted in the text.

15) Line 136: please write: mainly for those with chemoresistant and poor risk disease.

Response: The suggested sentence has been written in the text.

16) Line 159: mediastinal GCTs are not testicular GCT, so please delete the ‘T’ --> mediastinal GCT.

Response: Thank you for pointing that out. The letter ‘T’ has been deleted in the text.

17) Table 1 is good but gain of chromosome 12p is not a mutation.

Response: We apologize for our errors and the word ‘mutation’ has been replaced by ‘Genetic abnormalities’ on the Table 1.

Round 2

Reviewer 1 Report

no further comments